# In-situ measurements and modelling of the oxidation kinetics in films of a cooking aerosol proxy using a Quartz Crystal Microbalance with Dissipation monitoring (QCM-D)

Adam Milsom[1], Shaojun Qi[2], Ashmi Mishra[3], Thomas Berkemeier[3], Zhenyu Zhang[2], and Christian Pfrang[1,4]

[1]School of Geography, Earth and Environmental Sciences, University of Birmingham, Edgbaston, B15 2TT, Birmingham, UK.

[2]School of Chemical Engineering, University of Birmingham, Edgbaston, B15 2TT, Birmingham, UK.

[3]Multiphase Chemistry Department, Max Planck Institute for Chemistry, Hahn-Meitner-Weg 1, 55128 Mainz, Germany.

[4]Department of Meteorology, University of Reading, Whiteknights, Earley Gate, RG6 6BB, Reading, UK.

*Correspondence to*: Christian Pfrang (c.pfrang@bham.ac.uk)

**Abstract.** Aerosols and films are found in indoor and outdoor environments. How they interact with pollutants, such as ozone,
has a direct impact on our environment via cloud droplet formation and the chemical persistence of toxic aerosol constituents. The chemical reactivity of aerosol emissions is typically measured spectroscopically or by techniques such as mass spectrometry, directly monitoring the amount of material during a chemical reaction. We present a study which indirectly measures oxidation kinetics in a common cooking aerosol proxy using a low-cost Quartz Crystal Microbalance with Dissipation monitoring (QCM-D). We validated this approach by comparison with kinetics measured both spectroscopically
and with high-intensity synchrotron radiation. Using microscopy, we found that the film morphology changed and film rigidity increased during oxidation. There was evidence of surface crust formation on oxidised particles, though this was not consistent for all experiments. Crucially, our kinetic modelling of these experimental data confirmed that the oleic acid decay rate is in line with previous literature determinations, which demonstrates that performing such experiments on a QCM-D does not alter the underlying mechanism. There is clear potential to take this robust and low cost, but sensitive method to the field for in-situ
monitoring of reactions outdoors and indoors.

## 1 Introduction

Air quality is impacted by both natural and anthropogenic factors such as meteorology and cooking emissions(Chan and Yao, 2008; Huang et al., 2021), with cooking emissions estimated to contribute up to 10% of $PM_{2.5}$ in the UK (Ots et al., 2016). In

the West, people spend ~90% of their time indoors(Klepeis et al., 2001), and indoor air quality research has become important in recent years. Recent field studies have demonstrated the marked effect of processes such as cooking, cleaning and occupancy on indoor air quality in terms of particulate matter and volatile organic compound (VOC) emissions (Liu et al., 2021; Patel et al., 2020).

Surface films are present in the indoor environment and are formed by the deposition of particles and condensation of semi-volatile species with typical thicknesses in the order of a few hundred nanometres (Ault et al., 2020; Or et al., 2018). Indoor surface film chemistry is particularly important for air quality due to the high surface-to-volume ratio compared with outdoors. The composition of indoor films can vary between rooms and is influenced by the emission sources in each room (Or et al., 2018). For example, film samples collected in a kitchen after a stir-fry episode are likely to contain a larger amount of organic material including fatty acids (Or et al., 2020), which are major constituents of common cooking oils (Wang et al., 2020; Zahardis et al., 2006b).

Oleic acid is a major fatty acid component of cooking (Wang et al., 2020) and marine (Kirpes et al., 2019; Osterroht, 1993) organic emissions. As a surfactant, it can influence the cloud formation potential of aerosol particles, affecting the climate indirectly (Ovadnevaite et al., 2017). Oleic acid is also used as a marker for urban cooking emissions and the ratio of oleic acid to its saturated analogue (stearic acid) is a measure of how aged a sample of urban aerosols is (Wang et al., 2020). For these reasons, oleic acid is a common model system used to study heterogeneous reactions with oxidants such as ozone and $NO_3$ in the laboratory and with kinetic models (Berkemeier et al., 2021; Gallimore et al., 2017; King et al., 2004, 2009, 2020; Sebastiani et al., 2022; Shiraiwa et al., 2010, 2012; Woden et al., 2020; Zahardis and Petrucci, 2007). The atmospheric lifetime of oleic acid is longer than has been predicted in laboratory experiments (Robinson et al., 2006; Rudich, 2003), with recent evidence suggesting that the steric conformation of the fatty acid can impact on its chemical lifetime (Wang and Yu, 2021).

The viscosity of organic films and aerosols is an important factor in determining the rate at which heterogeneous processing occurs (i.e. the rates of water and reactive gas uptake) (Davies and Wilson, 2016; Koop et al., 2011; Shiraiwa et al., 2011). The ozonolysis of oleic acid is known to increase the viscosity (Hosny et al., 2016) and density(Katrib et al., 2005a) of the organic phase. An increase in viscosity decreases the rate of oxidative processing. Previous work has demonstrated that a viscous self-organised form of oleic acid (Milsom et al., 2021a, 2022a; Pfrang et al., 2017) reacts approximately an order of magnitude slower than the liquid form (Milsom et al., 2021b) and kinetic modelling of these results has shown that this could lengthen the chemical lifetime of oleic acid by several days under typical atmospheric conditions (Milsom et al., 2022c). There is a need for a technique that can measure both reaction kinetics and changes related to physical characteristics (i.e. viscosity) simultaneously with a high time resolution.

In this study, we used a quartz crystal microbalance with dissipation monitoring (QCM-D) to follow the reaction of oleic acid with ozone, which was complemented by white light interferometry (WLI) and Raman spectroscopy. Dissipation monitoring allowed us to infer changes in film rigidity and microscopic techniques revealed morphological changes during oxidation, including evidence for surface crust formation previously postulated and evidenced for this system (Milsom et al.,

2021b, 2022c). We derived kinetic decay constants from the QCM-D data and fitted a kinetic model to the Raman data to demonstrate the useful information that could be extracted from these experiments and to highlight the challenges associated

with this technique. We then drew atmospheric implications from our findings and suggest future directions for this experiment.

## 2 Methodology

Oleic acid (Part ref. 364525, technical grade 90%, Sigma Aldrich), methanol (ACS reagent, 99.8%) and oxygen gas (BOC, 99.5%) were used without further purification. Silicon dioxide coated QCM sensors (5 MHz, 14 mm diameter, Cr/Au/$SiO_2$

surface, Quartz Pro, Sweden) were rinsed with ethanol followed by a cleaning process in an oxygen plasma chamber (HPT-100, Henniker Plasma) at an oxygen flow rate of 10 sccm for 5 min prior to the deposition of oleic acid.

An oleic acid solution (10wt.% in methanol) was freshly prepared. The cleaned QCM sensor was placed on a spin coater (SPIN150i, APT GmbH) and spun at 6000 rpm as 60 μL oleic acid solution was added onto the sensor surface dropwise using a micro pipettor. Oleic acid coated sensors were tested the same day to avoid degradation due to the trace amount of ozone in

the ambient atmosphere.

      QCM-D works on the principle that the resonant frequency ($f$) of a piezoelectric quartz crystal can be monitored electronically. This $f$ decreases when small amounts of material are added to the quartz crystal. The dissipation factor ($D$) is a measure of the energy dissipated by the deposited material (Voinova et al., 1999). Both $f$ and $D$ are functions of the deposited film viscoelasticity. Generally, a lower $D$ implies a more rigid film.

The ozonolysis of oleic acid was studied using the coated sensors and a QCM-D (NEXT, openQCM, Italy), with which the frequency and energy dissipation history during the ozonolysis process was simultaneously recorded. We checked that $f$ was stable before starting ozone exposure experiments (Fig. S1) and how well $f$ and $D$ traces overlapped during the experiments (Fig. S2), with implications for the rigidity of the films discussed in sect. 3.1.

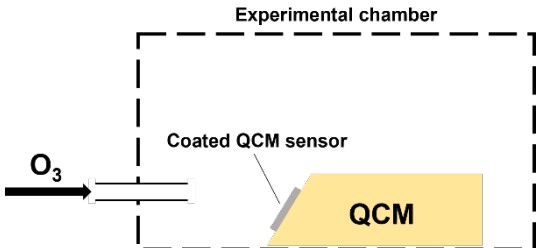

**Figure 1. A schematic illustration of the experiment presented in this study.**

      An experimental chamber made of polystyrene and with the inner walls lined with aluminium foil was used for ozonolysis experiments (Fig. 1). Ozone was produced by flowing oxygen at 1.2 L min$^{-1}$ through a commercial pen-ray ozoniser

(Ultraviolet Products Ltd, Cambridge, UK) which exposed the oxygen flow to UV radiation. The concentration of ozone was calibrated by UV-Visible spectroscopy and was determined to be $3.8 \pm 0.5$ ppm.

90        A white light interferometer (WLI, resolution 20×; MicroXAM2, KLA Tencor, California, U.S.A) was employed to establish the morphology of the oleic acid coated sensor. Scans were taken on each tested sensor at representative positions, i.e. within the test window (dia. 7 mm) that was previously subject to ozonolysis, at boundary of the test window, and sites far away from the test window. WLI data was analysed using an image processing program Gwyddion with which surface parameters, including surface roughness, 2-D/3-D height profiles could also be extracted. Raman features of the as-prepared

and the ozone-exposed oleic acid coatings were captured with a confocal Raman spectrometer (inVia™ by Renishaw, 20x optical magnification, laser wavelength 532 nm, laser power 10%). At least 30 accumulations were made to maximise the signal-to-noise ratio.

        Kinetic multi-layer models based on the Pöschl-Rudich-Ammann framework (Pöschl et al., 2007) are commonly used to analyse oleic acid ozonolysis experiments (Berkemeier et al., 2021; Milsom et al., 2022c, 2022b; Shiraiwa et al., 2010,

2012). The kinetic multilayer model of aerosol surface and bulk chemistry (KM-SUB) was employed to describe the reaction occurring between the deposited oleic acid and ozone (Shiraiwa et al., 2010). KM-SUB resolves processes such as gas adsorption and desorption, bulk diffusion, as well as surface and bulk chemistry. Although the deposited films were collections of smaller droplets, oleic acid was modelled as a flat film as the geometry was closer to that of a film for each individual droplet due to the high spreading ratio of oleic acid on the quartz surface. A KM-SUB model developed specifically for oleic

acid decay data measured by Raman spectroscopy, and optimised to 12 literature datasets, was fitted to the Raman data collected here (Berkemeier et al., 2021). A full description of the model is in Sect. S2 in the Supporting Information.



# 3 Results and Discussion

## 3.1 Kinetics of oleic acid ozonolysis

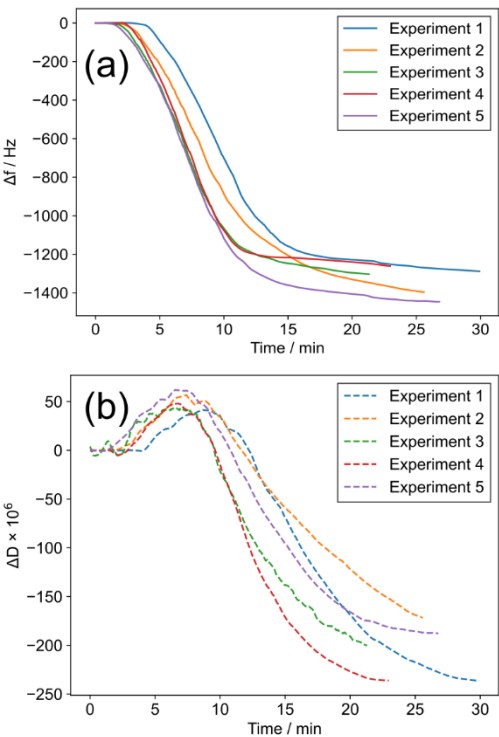

**Figure 2. (a) *Δf* vs time since the start of ozone exposure for five coatings. (b) *ΔD* vs time since the start of ozone exposure for five coatings, measured simultaneously to (a). The line colours link the same experiments in (a) and (b).**

We observed reproducible trends in *f* and *D* during ozone exposure (Fig. 2). After an initial build-up of ozone in the chamber, the resonant frequency shift (*Δf*) becomes more negative followed by a levelling off by the end of the experiment. If the Sauerbrey equation, which states that the mass per unit area deposited on a QCM crystal is inversely proportional to the crystal's measured resonant frequency (Demou et al., 2003), is valid then a decrease in *Δf* would mean in increase in mass per unit area on the crystal surface. However, inspection of the simultaneously monitored overtones suggests that the film is not rigid because they do not overlap entirely (Fig. S2). This is similar to the observation of Chao et al, who observed an increase in Δf during a solid-to-liquid phase transition even though the mass of their deposited samples increased whilst observing salt deliquescence (Chao et al., 2020). In our case, the decrease in Δf during oxidation does not necessarily mean the mass is increasing, as we expect some reaction products such as nonanal and nonanoic acid to be volatile (Müller et al., 2022; Zahardis and Petrucci, 2007). There is some evidence for a transition from a liquid to a solid-like state during ozonolysis: (i) we observe that ΔD is negative – more rigid films dissipate less energy; (ii) higher-molecular weight oligomeric compounds are known to form for this system during ozonolysis (Reynolds et al., 2006; Zahardis et al., 2006a); (iii) The condensed phase is known to

become denser during oxidation (Katrib et al., 2005a); (iv) we optically observed rigid structures formed on the surface of some particles after ozonolysis (see sect. 3.2).

Note that the f measured for these reactions is ~1200–1400 Hz lower than the original frequency. This is much higher than the stated standard deviation of 0.5 Hz quoted for $f$ measurements by the instrument manufacturer and suggests that much less reactive systems, or systems with a lower proportion of reactive material, could be studied.

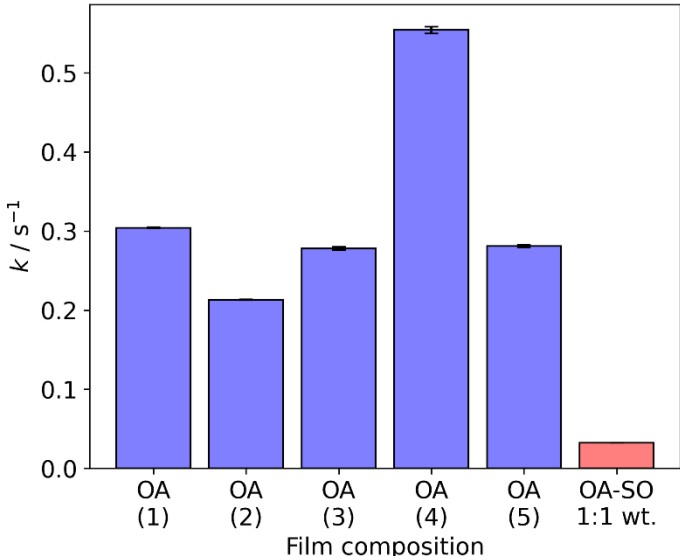


**Figure 3. The pseudo-first order decay constant ($k$) measured at the fastest point of each $\Delta f$ vs time plot presented in Fig. 2(a). The experiment numbers in brackets correspond with those presented in Fig. 2. The final bar is from the decay of a self-organised oleic acid-sodium oleate mixture analogous to previous work (Milsom et al., 2021b). All k values are measured at an ozone concentration of 3.8 ± 0.5 ppm. OA: oleic acid; OA-SO: oleic acid-sodium oleate mixture.**

The pseudo-first order decay constants ($k$) are generally consistent and variation is most likely due to slight variations in initial film thickness (Fig. 3). Taking the point at which the decay in $\Delta f$ is fastest returns a measure of the reaction kinetics. Although $\Delta f$ is not a measure of the amount of reactant remaining on the surface, applying pseudo-first order reaction kinetic analysis to the region of fastest $\Delta f$ decay can be used for comparisons with the same system (e.g. oleic acid) under different conditions. To test this, we coated a film of an oleic acid-sodium oleate (1:1 wt.) mixture and exposed it to the same oxidative
conditions as the pure oleic acid films. This mixture is known to self-organise into lamellar bilayers and is semi-solid (Milsom et al., 2021b). We found that $k$ for this viscous mixture was ~1 order of magnitude smaller than for the liquid oleic acid films presented here. This is due to the decreased diffusivity of ozone through the film and is consistent with the difference in reaction rates we have previously measured using X-ray scattering and Raman spectroscopy (Milsom et al., 2021b), validating this approach.


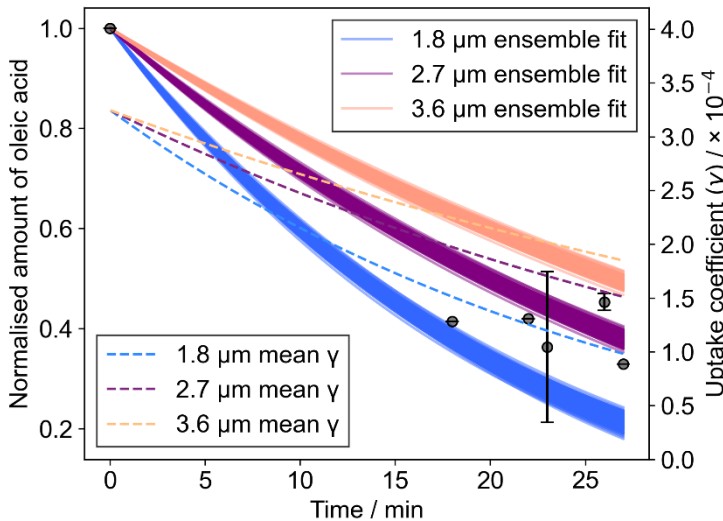

**Figure 4. The decay of oleic acid followed experimentally using the area of C=C Raman band at ~1650 cm$^{-1}$ normalised to the CH$_2$ band at ~1452 cm$^{-1}$. Ensemble outputs from a pre-optimised model of oleic acid ozonolysis (Berkemeier et al., 2021) are presented for a range of initial film thicknesses in the range measured by WLI. The mean uptake coefficient ($\gamma$) for each ensemble, derived from the model output, is also plotted.**

A kinetic model that has been pre-optimised to 12 unique datasets (Berkemeier et al., 2021) was applied to the experimental data measured by Raman spectroscopy (Fig. 4). Outputs from different model outputs with varying film thicknesses are presented as an ensemble of 167 optimised input parameter sets. The model also considers the formation of an oleic acid-Criegee intermediate adduct that would contribute to the carbon-carbon double bond signal observed in the Raman spectrum.

We found that the pre-optimised model fitted reasonably well to the experimental data when initialised at a thickness range determined by the range observed using WLI on films before ozone exposure (Fig. 4). The model does not describe changes in film morphology such as the coagulation of droplets into larger droplets, which was observed in the experiment (Fig. 5). Changes in the size of the deposited droplets will affect the uptake of ozone to oleic acid via changes to the surface area-to-volume ratio and the mixing time for ozone in the condensed phase (Pöschl et al., 2007). Therefore, a gradual increase of layer thickness will lead to a slowing of oleic acid consumption. We tested this hypothesis by splitting the model into 5 distinct time periods, each new period resulting in film thickening (Fig. S3 in the Supporting information). However, time-resolved morphological information would be required to constrain this particular feature. There could also be an effect from surface crust formation, slowing the reaction (Milsom et al., 2021a, 2022c; Pfrang et al., 2011). Though the exact kinetic effect

of crust formation and morphology change cannot be deconvoluted here, we believe that the significant change in morphology (i.e. increase in average film size) dominates.

It is possible to extract an uptake coefficient for ozone ($\gamma$) from the output of the KM-SUB model (Shiraiwa et al., 2010). In this case, $\gamma$ is the fraction of ozone molecules that collide with the oleic acid surface taken up by oleic acid. The values of $\gamma$ varied from $\sim 3 \times 10^{-4}$ to $\sim 1 \times 10^{-4}$ as the reaction proceeded (Fig. 4). This is within the range that has been calculated using resistor-based analytical models for oleic acid (in the order of $\sim 3.4 \times 10^{-4}$–$7.5 \times 10^{-4}$) (Hearn and Smith, 2004; Nash et al., 2006).The trend of a decreasing $\gamma$ as a result of oxidation is consistent with previous work (Mendez et al., 2014). It is expected that these uptake values are an upper limit for what would be the case in the atmosphere. Particles of oleic acid mixed with other components such as stearic acid (the C18 saturated analogue of oleic acid) (Katrib et al., 2005b), C17 and C16 fatty acids (Ziemann, 2005) generally have a lower calculated uptake coefficient than pure oleic acid particles in those respective studies.

## 3.2 Morphology changes

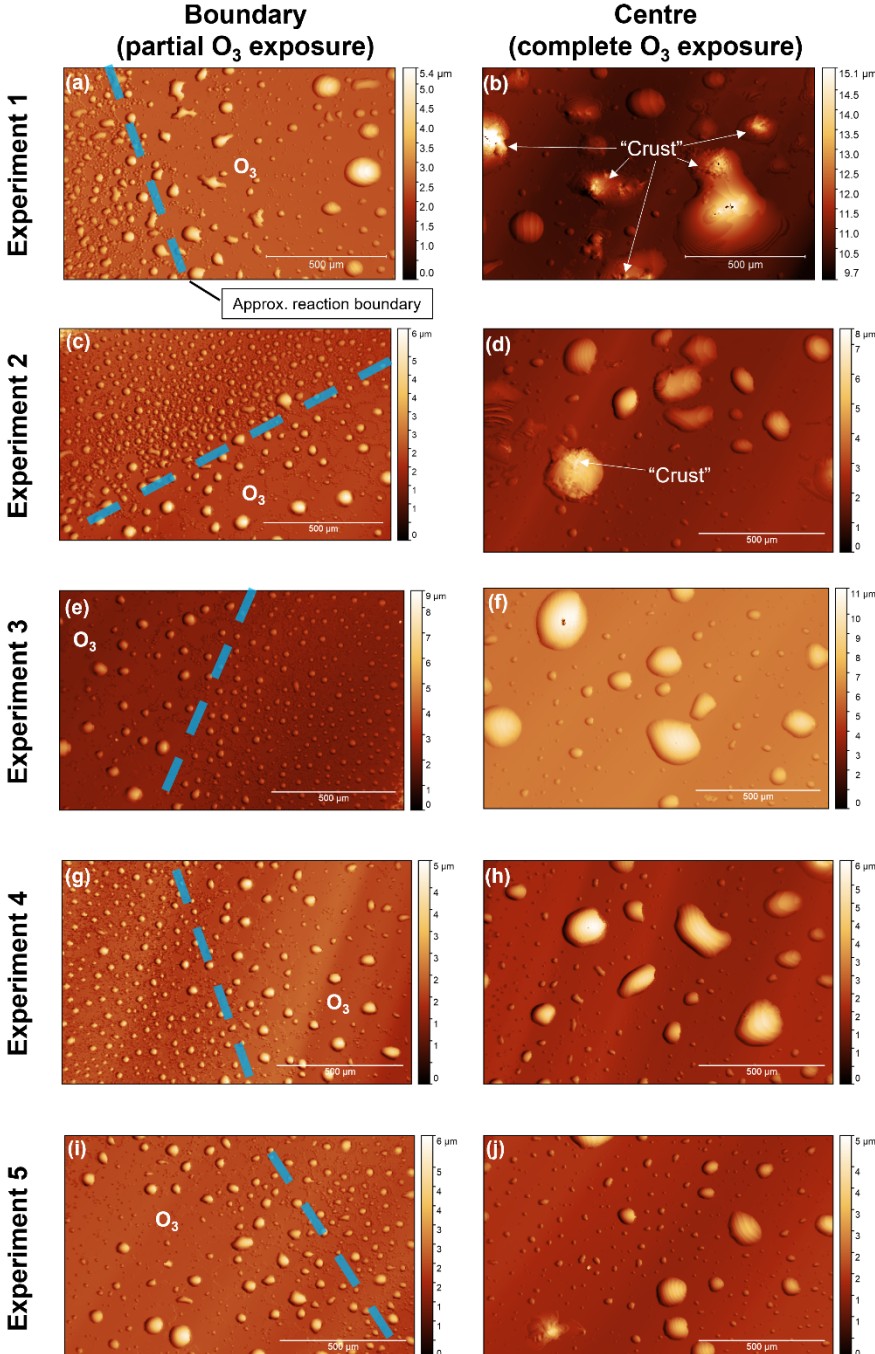

**Figure 5. White light interferometry images of oleic acid coated QCM crystal surfaces for five separate ozonolysis experiments. Images were taken at the reaction boundary i.e. the outer region where the surface was not exposed to the oxygen-ozone mixture**

due to the design of the QCM crystal holder (the diameters of the crystal and the test window are 14 and 7 mm, respectively). The approximate location of the reaction boundary is illustrated. Images at the centre of the QCM crystal surface show particles fully exposed to the oxygen-ozone mixture. Observations of a surface "crust" forming are labelled in panels (b) and (d). The approximate height of each droplet is indicated by the heatmap.

There is a difference in morphology between unoxidised and oxidised particles, with oxidation appearing to cause droplet coagulation (Fig. 5). The design of the QCM instrument, where the sample holder window had a smaller diameter than the QCM crystal, meant that only the central part of the QCM surface was exposed to ozone, with the outer regions of the surface not exposed to the chamber environment. This allowed us to image the boundary between these two regions and compare oxidised and unoxidised droplets. Initial droplet heights were mostly in the region of $\sim 1 - 9$ μm with a mean height calculated as $\sim 2$ μm. After oxidation, coagulation occurred resulting in fewer, larger droplets with maximum heights of $\sim 10 - 15$ μm (see sect. S4 for the representative height scans used for this analysis).

In addition to droplet coagulation, we observed microscopic evidence of a crust forming, triggered by film oxidation (Fig. 5). This was not consistent for all droplets, however some oxidised particles have clear rough patches, which we have defined as a crust, on their surfaces as compared to the relatively smooth liquid surfaces of other particles (labelled in Fig. 5(b) & (d)). There has been previous experimental and modelling evidence that crusts could form on the surface of oxidising oleic acid films and particles (Milsom et al., 2021a, 2021b, 2022c). Similar morphological changes have been observed by optical microscopy and atomic force microscopy (Hung and Tang, 2010; Liu et al., 2020). Here, WLI has confirmed the reproducible nature of these morphology changes along with a more quantitative description of the particle size changes observed.

## 3.3 Atmospheric implications

We have demonstrated that the phase state of deposited oleic acid changes during ozonolysis. An increase in viscosity has been monitored before for the oleic acid-ozone system (Hosny et al., 2016). However, this involved adding a fluorescent probe molecule to the sample. Here, we confirm with the non-invasive QCM-D experiment that the deposited cooking aerosol proxy becomes more rigid during ozonolysis.

A viscous layer coating aerosol material is thought to contribute to the persistence of pollutants in the atmosphere (Mu et al., 2018; Shrivastava et al., 2017). In this study, we have qualitatively observed a crust forming on the outside of oxidised oleic acid particles. We have previously observed a surface layer of aggregates forming during the ozonolysis of oleic acid-sodium oleate particles using X-ray scattering, which we assumed were high molecular weight products (Milsom et al., 2021a). Modelling of the oleic acid-sodium oleate system also suggests a crust could form (Milsom et al., 2022c). Our microscopic evidence presented here was not consistent for all experiments. However, it does add to the growing body of evidence for crust formation, potentially increasing persistence of atmospheric pollutants co-emitted with oleic acid.

## Conclusions

The high temporal resolution of QCM method presented here has allowed us to establish a measure of the ozonolysis kinetics of a commonly studied cooking aerosol proxy. We have confirmed that the relative decay rate of oleic acid compared to a viscous form of oleic acid, measured using the QCM method, agrees with that derived using X-ray scattering and Raman spectroscopy (Milsom et al., 2021b). An analysis using a kinetic model, pre-optimised to 12 oleic acid ozonolysis datasets, demonstrates that the oleic acid decay rate measured with the QCM method are consistent with previous experiments on aerosol particles in the literature (Berkemeier et al., 2021).

We can now qualitatively follow the rigidity, or phase state, of these oxidising films over time using the dissipation measured by the QCM-D instrument. For films of uniform thickness, there is the possibility of applying models of viscoelasticity to QCM-D data to derive the viscosity of coated films (Voinova et al., 1999). Future work should focus on this as a potential real-time measure of the viscosity of environmental films.

The portable QCM-D experiment described here could be used in the field to follow the kinetics of the interaction of real environmental films with pollutants (e.g. ozone and $NO_2$). Similar experiments have been carried out using a QCM regarding the water uptake of deposited films in the context of air quality and atmospheric chemistry (Asad et al., 2004; Demou et al., 2003; Schwartz-Narbonne and Donaldson, 2019).

## Acknowledgements

This work was supported by NERC grant number NE/T00732X/1 and a NERC Discipline Hopping Grant number 1002711. A. Mishra was supported by the Max Planck Graduate Center with the Johannes Gutenberg-Universität Mainz (MPGC).

**Author contributions.** AMilsom wrote the initial draft of the manuscript, carried out QCM-D experiments, processed and analysed the data. SQ carried out QCM-D experiments, collected Raman data and contributed to the manuscript. AMishra carried out the KM-SUB kinetic modelling and contributed to the manuscript. TB provided guidance and analytical input to the kinetic modelling; contributed to the manuscript. ZZ provided access to the QCM-D, helped interpret the results and contributed to the manuscript. CP secured the main funding (NERC grant number NE/T00732X/1), guided the experimental setup, helped interpret the results and contributed to the manuscript.

**Data availability.** Experimental and modelling data are available at https://doi.org/10.5281/zenodo.8296882

## Competing interests

At least one of the (co-)authors is a member of the editorial board of Atmospheric Chemistry and Physics.

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
