# Peer review of "Technical note: in-situ measurements and modelling of the oxidation kinetics in films of a cooking aerosol proxy using a Quartz Crystal Microbalance with Dissipation monitoring (QCM-D)"

_EGUsphere, 2023_

## Author Response (AR1)

**Authors' response to the reviews of "Technical note: in-situ measurements and modelling of the oxidation kinetics in films of a cooking aerosol proxy using a Quartz Crystal Microbalance with Dissipation monitoring (QCM-D)"**

Authors' responses (blue). Manuscript text edits (red).

**Reviewer 1**

Milsom et al. characterized the influence of ozone exposure on oleic acid films using quartz crystal microbalance (QCM), Raman spectroscopy, and white light interferometry (WLI) techniques. From the measured decreases in resonant frequency and dissipation factor of the oleic-acid-coated QCM sensors, the authors infer that oleic acid was oxidized and the droplet density increased; in separate experiments, the decay rate of oleic acid droplets was faster than that of mixed oleic acid / sodium oleate droplets. They observed evidence of coagulation and crust formation from the WLI sample images. The combination of techniques used here is proposed as a low-cost, field-deployable method for the measurement of the oxidation kinetics of other compounds.

**General Comments**

In complex/ambient samples (in the authors' terminology, "real environmental films", L220), the level of unsaturation and the corresponding ozone reactivity are likely to be much lower than those containing >50% oleic acid as studied here. Reducing the oleic acid fraction from 100% to 50% already reduced the ozone reactivity by a factor of 10 (Figure 3). What is the measured change in resonant frequency / dissipation factor of, for example, films containing 10%, 1%, and 0.1% oleic acid mixtures? How about for bare QCM sensors with no added films? There is also no discussion or method demonstration of controlled exposure of films to, for example, hydroxyl radicals which are a much less selective oxidant than ozone. It is not clear to me what the lowest sample ozone reactivity is that can be meaningfully measured with this technique, and because of that, its potential application to the measurement of oxidation kinetics in samples that don't have significant levels of unsaturation (C=C bonds) seems limited or at best unknown.

- **Using different relative amounts of oleic acid.** We thank the reviewer for this comment. There is a slight misunderstanding here. There is a double bond in the sodium oleate that is in the oleic acid-sodium oleate mixture. So it is still the case that the C=C fraction is 100%. The kinetic effect is due to the much greater viscosity in the oleic acid-sodium oleate mixture compared with liquid oleic acid, limiting the reaction rate. We have made a minor adjustment to the text to spell this effect out:

"We found that k for this viscous mixture was ~1 order of magnitude smaller than for the liquid oleic acid films presented here. This is due to the decreased diffusivity of ozone through the film and is consistent with the difference in reaction rates we have previously measured using X-ray scattering and Raman spectroscopy (Milsom et al., 2021b), validating this approach."

- **The sensitivity of the QCM-D instrument.** The reviewer raises a valid point. The change in frequency we measured for these films was in the order of 1200-1400 Hz. The manufacturer quotes a standard deviation for the instrument as 0.5 Hz. If we take a conservative estimate that the limit of detection is 2 Hz, this technique is very capable of measuring much smaller changes in the deposited mass. We have included an additional couple of sentences at the end of the first paragraph of section 3.1:

  "Note that the f measured for these reactions is ~1200–1400 Hz lower than the original frequency. This is much higher than the stated standard deviation of 0.5 Hz quoted for f measurements by the instrument manufacturer and suggests that much less reactive systems, or systems with a lower proportion of reactive material, could be studied."

- **Control experiments.** We thank the reviewer for raising this. The data presented in Fig. 2 are datapoints measured after ozone exposure was initiated. In the initial 1–2 minutes of the experiment while ozone was beginning to fill the reaction chamber, $\Delta f$ remained around 0 Hz. We have now included an example of the raw f vs time data in the supporting information for one of the experiments carried out which includes the period before we started ozone exposure:

[Figure]

We have also referred to this in the main text in the methodology section:

"We checked that f was stable before starting ozone exposure experiments (Fig. S1) and how well f and D traces overlapped during the experiments (Fig. S2), with implications for the rigidity of the films discussed in sect. 3.1."

- **Reactions with other reactive gasses.** The reviewer makes a valid point that reactivity with less selective gases such as OH radicals would be interesting. The QCM-D measures changes in the fundamental frequency of the crystal (related to mass and rigidity) and is not sensitive to the specific functional groups (i.e. C=C bonds) in the film. We suggest that any reaction which results in a change in surface film properties, such as mass, rigidity and density, would be observable and the relative kinetic decay rates obtainable in a similar way to what we have presented for the oleic acid-ozone system. Reactions with OH radicals are possible. However, quantifying OH radical exposure is difficult and is beyond the scope of this technical note.

In my opinion, the scope of the kinetic analysis is also limited. It seems like it should be possible to calculate additional - and arguably more useful – kinetic data products than the first-order loss rate of oleic acid; for example, I think the reactive uptake coefficient of ozone could be calculated from the first-order decay rate observed with QCM, the droplet sizes obtained from WLI samples, and the KM-SUB model. Please calculate the corresponding uptake coefficients for the results shown in Figures 2 and 3 and discuss these values in the context of previously measured ozone uptake coefficient values for pure and mixed oleic acid aerosols. Also, discuss any associated limitations in the technique that may prevent accurate retrieval of these values, as applicable.

- **More detailed kinetic analysis.** We agree with the reviewer that a more detailed kinetic analysis would strengthen this manuscript. The classic analytical methods of calculating an uptake coefficient for oleic acid rely on using a measure of oleic acid concentration, which QCM-D does not provide. We have therefore focussed this analysis on the KM-SUB model outputs, which provide time-resolved uptake coefficients. We have included these outputs in an updated Fig. 4. Uptake coefficients are slightly lower than what has been measured for the ozonolysis of oleic acid (Hearn and Smith, 2004), but is within the same order of magnitude. The updated Fig. 4 is displayed below:

[Figure]

An extra paragraph of discussion has been added to section 3.1:

"It is possible to extract an uptake coefficient for ozone ($\gamma$) from the output of the KM-SUB model (Shiraiwa et al., 2010). In this case, $\gamma$ is the fraction of ozone molecules that collide with the oleic acid surface taken up by oleic acid. The values of $\gamma$ varied from ~$3 \times 10^{-4}$ to ~$1 \times 10^{-4}$ as the reaction proceeded (Fig. 4). This is within the range that has been calculated using resistor-based analytical models for oleic acid (in the order of ~$3.4 \times 10^{-4}$–$7.5 \times 10^{-4}$) (Hearn and Smith, 2004; Nash et al., 2006).The trend of a decreasing $\gamma$ as a result of oxidation is consistent with previous work (Mendez et al., 2014). It is expected that these uptake values are an upper limit for what would be the case in the atmosphere. Particles of oleic acid mixed with other components such as stearic acid (the C18 saturated analogue of oleic acid) (Katrib et al., 2005b), C17 and C16 fatty acids (Ziemann, 2005) generally have a lower calculated uptake coefficient than pure oleic acid particles in those respective studies."

**Minor/Technical Comments**

L56 – Suggest changing "by an order of days" to "by several days" or similar

We have made this amendment.

L104 – In "the sect. S1", delete "the" and capitalize "Sect."

We have made this amendment.

L130 – Suggest deleting "Only", add "nonanal and nonanoic acid are the only oleic acid ozonolysis products known…"

This section was changed in response to reviewer 2 (see responses below).

L166 – change "A" to lower case

We have made this amendment.

L189 – in the context of this discussion/figure, what are the specific morphological features define a "crust"?

We have adjusted this description slightly to spell out that the smooth surfaces correspond to a liquid (oleic acid) and the rough patches correspond to a crust:

"…This was not consistent for all droplets, however some oxidised particles have clear rough patches, which we have defined as a crust, on their surfaces as compared to the relatively smooth liquid surfaces of other particles (labelled in Fig. 5(b) & (d))…"

L191 - It would be useful if a label/annotation could be added to Figures 5b and 5d to indicate where the authors think a crust formed - to the untrained eye, this may not necessarily be obvious.

We thank the reviewer for this suggestion. We have now annotated where we believe the crust has formed in those panels of the figure. See the updated Fig. 5 below:

[Figure]

The manuscript by Milsom et al. presents kinetic measurements of oleic acid ozonolysis using a low-cost quartz-crystal microbalance with dissipation (QCM-D). The authors observed decreases in both resonant frequency and dissipation. The retrieved first-order reaction rate agrees well with Raman spectroscopy data. Overall, I think this is an interesting study, which well demonstrates that the low-cost QCM-D instrument can be used to investigate atmospherically relevant multiphase reaction kinetics. However, I do have a few concerns about the interpretation of the results.

Line 122-128: "This observation suggests an increase in mass per unit area on the QCM crystal surface via the Sauerbrey equation, which states that the mass per unit area deposited on a QCM crystal is inversely proportional to the crystal's measured resonant frequency…

The apparent increase in mass per unit area observed during ozonolysis could be due to an increase in film density. This has been observed previously for oleic acid ozonolysis, where density increases from 0.89 to 1.12 g cm-3 with increasing ozone exposure, presumably due to ozonolysis products having higher densities…"

My concern is that the initial film of the liquid oleic acid film may have a large dissipation, therefore the QCM frequency does not follow the linear relationship of the Sauerbrey equation. In this case, the decrease in resonant frequency does not necessarily mean an increase in mass per unit area. For example, Chao et al. (ACS Omega, 2020) reported that the resonant frequency of QCM increases during the solid-to-liquid phase transition induced by the deliquescence of salts, although the actual mass largely increases. It seems to me that the QCM-D results reported in this study behave just like the reverse process. I think the phase transition from liquid to solid-like state during oxidation played a major role in driving both f and D changes, and the density change only had a small effect.

We thank the reviewer for this very helpful comment. We believe this has strengthened the interpretation of our results. We had highlighted the possibility of the film density explaining the trends in *f* and *D.* However, upon closer inspection of the overtones measured by the QCM-D instrument, the *f* and *D* curves do not quite overlap for all overtones. This suggests that the film is not rigid and the observations pointed out by the reviewer are probably similar to Chao et al. (who we have now referenced in our improved manuscript). See the figure below, which we have added to the Supporting Information. We have plotted the normalised values for ease of comparison. These have not been normalised to the overtone number, so there will be differences in the absolute frequency change for each overtone (*n*).

We refer to this as Fig. S2 with updated text presented in our response to reviewer 1.

[Figure]

We have also updated the discussion in the main text, section 3.1:

"…If the Sauerbrey equation, which states that the mass per unit area deposited on a QCM crystal is inversely proportional to the crystal's measured resonant frequency (Demou et al., 2003), is valid then a decrease in $\Delta f$ would mean in increase in mass per unit area on the crystal surface. However, inspection of the simultaneously monitored overtones suggests that the film is not rigid because they do not overlap entirely (Fig. S2). This is similar to the observation of Chao et al, who observed an increase in $\Delta f$ during a solid-to-liquid phase transition even though the mass of their deposited samples increased whilst observing salt deliquescence (Chao et al., 2020). In our case, the decrease in $\Delta f$ during oxidation does not necessarily mean the mass is increasing, as we expect some reaction products such as nonanal and nonanoic acid to be volatile (Muller et al., 2022; Zahardis and Petrucci, 2007). There is some evidence for a transition from a liquid to a solid-like state during ozonolysis: (i) we observe that $\Delta D$ is negative – more rigid films dissipate less energy; (ii) higher-molecular weight oligomeric compounds are known to form for this system during ozonolysis (Reynolds et al, 2006; Zahardis et al., 2006a); (iii) The condensed phase is known to become denser during oxidation (Katrib et al., 2005a); (iv) we optically observed rigid structures formed on the surface of some particles after ozonolysis (see sect. 3.2)."

In addition, the authors should consider adding baseline measurements for bare sensors, and report delta D and delta F data relative to the baseline. The data should not be interpreted with Sauerbrey equation if the D value is large relative to the baseline. Also, reporting data from different overtones can help to verify of the Sauerbrey equation is applicable. Alternatively, the authors could try measurements with lower sample mass, such

that the Sauerbrey equation is valid and the mass change during oxidation can be determined.

We thank the reviewer for this comment. This is mostly addressed in our response to reviewer 1 concerning the stability of *f* before starting the experiment. ΔD and Δf are already measured relative to the baseline, which we determined from the average of the *f* and *d* values before ozone exposure (see the new Fig. S1 and S2).

In our previous response, we adjusted our interpretation to suggest that the Sauerbrey equation is probably not the best way to describe the data. None of our analyses had used the Sauerbrey equation anyway. We had originally signposted it as a way of justifying our observations.

We agree that experiments on thinner films would allow us to use the Sauerbrey equation to determine mass changes. However, we are interested in the kinetics in this study – no matter the direction of the trends in *Δf* and *ΔD*, the trend in the kinetics derived from these measurements are consistent with those measured by X-ray scattering and Raman spectroscopy (which we have signposted in the text).

**References**

Hearn, J. D. and Smith, G. D.: Kinetics and product studies for ozonolysis reactions of organic particles using aerosol CIMS, J. Phys. Chem. A, 108(45), 10019–10029, doi:10.1021/jp0404145, 2004.

Chao, H. J., Huang, W. C., Chen, C. L., Chou, C. C. K. and Hung, H. M.: Water Adsorption vs Phase Transition of Aerosols Monitored by a Quartz Crystal Microbalance, ACS Omega, 5(49), 31858–31866, doi:10.1021/acsomega.0c04698, 2020.